# Involvement of Potassium Channel Signalling in Migraine Pathophysiology

**DOI:** 10.3390/ph16030438

**Published:** 2023-03-14

**Authors:** Mohammad Al-Mahdi Al-Karagholi

**Affiliations:** Danish Headache Center, Department of Neurology, Rigshospitalet Glostrup, Faculty of Health and Medical Sciences, University of Copenhagen, 1165 Copenhagen, Denmark; mahdi.alkaragholi@gmail.com

**Keywords:** aura, K_ATP_ channels, BK_Ca_ channels, ion channels, headache

## Abstract

Migraine is a primary headache disorder ranked as the leading cause of years lived with disability among individuals younger than 50 years. The aetiology of migraine is complex and might involve several molecules of different signalling pathways. Emerging evidence implicates potassium channels, predominantly ATP-sensitive potassium (K_ATP_) channels and large (big) calcium-sensitive potassium (BK_Ca_) channels in migraine attack initiation. Basic neuroscience revealed that stimulation of potassium channels activated and sensitized trigeminovascular neurons. Clinical trials showed that administration of potassium channel openers caused headache and migraine attack associated with dilation of cephalic arteries. The present review highlights the molecular structure and physiological function of K_ATP_ and BK_Ca_ channels, presents recent insights into the role of potassium channels in migraine pathophysiology, and discusses possible complementary effects and interdependence of potassium channels in migraine attack initiation.

## 1. Introduction

Migraine is a primary headache disorder affecting more than 15% of the global adult population in their most productive years of life with a health and economic burden of billions of dollars globally [1,2,3]. The clinical manifestation of migraine is recurrent attacks with a severe and usually unilateral and throbbing headache, lasting 4–72 h and associated with nausea and/or light and sound sensitivity [4]. In one-third of individuals with migraine, the headache phase is preceded by transient focal neurological disturbances, the so-called migraine aura phase, whose underlying mechanism is considered to be cortical spreading depression (CSD) [5,6].

The importance of ion channels in the pathogenesis of migraine has gathered considerable attention in the past three decades [7,8,9]. Altered ion channel function causes a range of neurological diseases known as *channelopathies*, such as epilepsy and episodic ataxia [10]. Due to disturbances of neurological function, the phenotype of channelopathies is paroxysmal symptoms [11]. Ion channels are expressed in cranial arteries and trigeminal afferents, where they essentially regulate vascular tone and signal transmission in the cephalic pain system [12,13,14]. Genetic studies investigating mechanistic insights underlying migraine subphenotypes revealed mutations in genes encoding the α1 subunit of the CaV2.1 P/Q-type voltage-gated Ca^2+^ channel (*CACNA1A*) and the α1 subunit of the neuronal NaV1.1 voltage-gated Na^+^ channel (*SCN1A*), respectively [15,16]. Furthermore, endogenous signalling molecules involved in migraine including calcitonin gene-related peptide (CGRP) and pituitary adenylate cyclase-activating polypeptides (PACAPs) are dependent on ion channel activation, particularly potassium channels [17,18]. A series of intervention studies implicated ATP-sensitive potassium (K_ATP_) channels and large (big) calcium-sensitive potassium (BK_Ca_) channels in migraine pathogenesis (Figure 1). K_ATP_ and BK_Ca_ channels belong to a large family of voltage- and ligand-gated potassium channels. These channels are normally closed at resting membrane potentials but open rapidly upon depolarization, accounting for a large part of the repolarization phase. The present review outlines the biochemical identities and structures of K_ATP_ and BK_Ca_ channels, summarizes recent mechanistic insights into their role in migraine pathophysiology, and discusses potential complementary effects and interdependence of potassium channels in migraine attack initiation.

## 2. Methods

No protocol was registered for this narrative review. References for the present review were identified by a narrative search of the **PubMed database** regarding potassium channels and migraine on 27 November 2022. Following search terms “K_ATP_
*channel AND migraine*”, “BK_Ca_
*channel AND migraine*”, and “*Potassium channels AND Headache AND Migraine*” were used. There were no restrictions in terms of the language or date of publication. Additionally, references from relevant articles were identified. The final reference list was generated based on relevance to the topic by reading the title and abstract.

## 3. ATP-Sensitive Potassium (K_ATP_) Channels

In the late 1980s, there was an extraordinary interest in targeting the K_ATP_ channel for the treatment of asthma, angina pectoris, and hypertension [10], and several K_ATP_ channel openers (KCO) such as levcromakalim, nicorandil, and pinacidil have been developed. Remarkably, clinical trials assessing pharmacodynamical properties of KCO reported headache as a frequent adverse event [10]. Preclinical studies showed that KCO dilated cranial arteries and induced hypersensitivity in a mouse model of provoked migraine-like pain, and the K_ATP_ channel blocker glibenclamide attenuated dilation and completely blocked trigeminal pain transmission [19,20]. In an experimental human model, intravenous infusion of levcromakalim triggered headache associated with dilation of cranial arteries in healthy participants [21]. Additionally, all patients with migraine developed migraine attacks after levcromakalim infusion, and patients with migraine with aura reported migraine aura upon levcromakalim infusion [22]. Thus, levcromakalim is the most powerful migraine trigger ever tested in human and the first trigger of migraine aura. These remarkable preclinical and clinical observations are informing hypotheses about potential molecular mechanisms of action that require elucidation to realize the full potential of K_ATP_ channels in the treatment of migraine. In particular, despite compelling evidence that activation of K_ATP_ channels is a critical mediator of migraine attack initiation, uncertainty remains about where in the trigeminovascular system (TVS) [23] and at what level in signal transduction pathways targeting K_ATP_ channels could have therapeutic effects and to what degree they can be isolated by developing novel chemical probes with differing specificity and selectivity profiles.

Several tissues express K_ATP_ channels including cells within the peripheral and central nervous system, cardiac myocytes, and pancreatic cells [24,25]. K_ATP_ channels are composed of eight subunits (octameric complex) belonging to two structurally and functionally distinct protein families [26]. Four pore-forming subunits belong to the inward rectifier potassium (Kir) channel family and four regulatory sulfonylurea receptor (SUR) subunits belong to the ATP-binding cassette (ABC) transporter family (Figure 2) [27]. Six subfamilies of the Kir channel have been identified, and the Kir subfamily detected in K_ATP_ channels is the Kir6 subunit. The Kir6 subunit is expressed in two isoforms, Kir6.1 and Kir6.2, transcribed from two different genes, KCNJ8 and KCNJ11, respectively [28]. Seven subfamilies of the ABC transporter family have been identified (ABC*A*-ABC*G*), and the SUR subunit belongs to the ABC*C* subfamily [29,30]. The SUR subunit exists in three isoforms: SUR1, SUR2A, and SUR2B. SUR1 is transcribed from the ABCC8 gene, whereas SUR2A and SUR2B are splice variants encoded from the same gene, ABCC9 [27,28]. The latter 2 vary in 42 amino acid residues in their distal COOH-terminal (C42), which gives them physiologically distinguishable qualities [31]. By acting as sensors of the intracellular ATP:MgADP ratio, K_ATP_ channels connect the metabolic state of the cell to the membrane potential in response to extracellular and intracellular changes, such as hypoxia, ischemia, or hypo- and hyperglycemia [31]. An increase in cAMP or cGMP levels or a decrease in intracellular ATP activates (opens) K_ATP_ channels, causing potassium efflux and membrane hyperpolarization, which depending on the tissue will lead to a specific cellular response [31]. In smooth muscle cells, for instance, K_ATP_ channel activation decreases the opening probability of voltage-gated Ca^2+^ channels (VOCC) and leads to vasodilation by reducing the cytosolic Ca^2+^ concentration [32].

Levcromakalim is a selective SUR2B-K_ATP_ channel opener (Figure 3), and the commonest K_ATP_ channel subunit expressed in the TVS is SUR2B [33,34,35,36]. Accordingly, SUR2B subunit emerges as a potential therapeutic drug target for the treatment of migraine. However, a selective SUR2B blocker is not available. The anti-diabetic drug glibenclamide is a nonselective SUR blocker with higher affinity to SUR1 subunit expressed in pancreas, and thus, hypoglycaemia is a frequent side effect after glibenclamide administration [37]. Additionally, a series of intervention studies reported that glibenclamide had no effect on the triggered headache after CGRP, PACAP38, or levcromakalim in healthy participants [38,39,40,41].

## 4. High-Conductance (Big) Calcium-Activated Potassium (BK) Channels

Calcium-activated potassium (BK_Ca_) channels, also called Slo1 family channels, were identified when a prominent outward K^+^ current was discovered upon membrane depolarization and/or after an influx of Ca^2+^ [42]. Of all K^+^ selective channels, BK_Ca_ channels have the largest single-channel conductance and consist of two distinct regions with segments S0–S10. The *core region* contains segments S0–S6 which resemble a canonical voltage-gated K^+^ channel and a large intracellular carboxyl extension including segments S7–S10 [43]. The distal part of the carboxyl region (S9–S10), termed the *tail region*, includes a highly conserved domain among Slo1 proteins from different species, termed the calcium bowl (Figure 4) [44]. Auxiliary β-subunits interact with α-subunits to form a non-covalent BK_Ca_ channel complex. Four distinct β-subunits (β1-β4) have been discovered [45]. The β2 and β3 subunits share sequence similarities with β1, but unlike β1 and β4 which favour the active conformation, β2 and β3 promote a fast-inactive conformation in BK_Ca_ channels [46]. The β1 subunit is expressed primarily in smooth muscle and some neurons [47], while the β4 subunit is highly expressed in the brain [48].

Apart from sensitivity to depolarization and intracellular Ca^2+^, BK_Ca_ channels are directly regulated by an imbalance between cellular kinase and phosphatase enzymes. Numerous common serine/threonine kinases, including PKA, PKG, and diacylgycerol/Ca^2+^-dependent protein kinase C (PKC) modulate BK_Ca_ channel activity, but the PKA phosphorylation is possibly the best-understood mechanism [45,46]. Phosphorylation occurs near the C-terminal edge of the calcium-bowl sequence, and the open-channel probability increases when all four subunits of a homomeric BK_Ca_ channel are phosphorylated (Figure 4).

Several findings indicate a possible role of BK_Ca_ channels in migraine. Firstly, β1-subunit BK_Ca_ channels are expressed in the TVS including smooth muscle cells in cranial arteries, TG and TNC (Figure 5) [8,49]. Secondly, BK_Ca_ channels are activated by cAMP-PKA and cGMP-PKG [50,51]. Thirdly, the infusion of the BK_Ca_ channel opener MaxiPost triggers headache in healthy volunteers [52]. Lastly and most importantly, a recent study showed that patients with migraine developed migraine attacks after MaxiPost infusion [53]. Collectively, these data provide a strong mechanistic rationale to identify a synergistic or additive treatment effect by targeting BK_Ca_ channels. Several BK_Ca_ channel blockers including iberiotoxin, paxilline, and charybdotoxin have been used preclinically to inhibit the physiological effects induced by CGRP and PACAP [12,54]. However, these blockers are non-selective and not approved for clinical use (Figure 6). More selective blockers to the auxiliary β1-subunit, which is highly expressed in the trigeminovascular system, would be useful as a candidate for future migraine therapies.

## 5. Potassium Channel Interplay in Migraine Pathophysiology

Ion channel interaction is a well-known phenomenon, and several ion channels share intracellular signalling cascades despite exhibiting different functions. In order to discuss the interplay between presented ion channels, the distinctive location of these channels must be taken as a starting point. Nociceptive and non-nociceptive signals from the meninges and other cranial tissues reach multiple cortical areas through a sensory tract consisting of peripheral trigeminovascular neurons in the trigeminal ganglion (TG), central trigeminovascular neurons in the trigeminal nucleus caudalis (TNC), and thalamic neurons. The following section focuses on potassium channel expression and interplay in (1) dural afferents and smooth muscle cells in cranial vessels, (2) the TG, and (3) TNC. Figure 7 illustrates a possible ion channel interplay between acid-sensing ion channel (ASIC), BK_Ca_ channel, K_ATP_ channel, N-methyl D-aspartate receptor (NMDAR) [55], transient receptor potential channels (TRPA1, TRPM8, TRPV1, and TRPV4) based on their presumed occurrence in the trigeminovascular system and molecular functions.

### 5.1. Trigeminal Afferents and Neurovascular Smooth Muscle Cells

Trigeminal afferents are thinly myelinated Aδ-fibers or unmyelinated C-fibers expressing numerous ion channels which allow passage of cations, importantly Ca^2+^. Antidromic conduction and Ca^2+^ influx elicits CGRP release from C-fibres (first order neurons) to blood vessel walls causing activation of K_ATP_ and BK_Ca_ channels in neurovascular smooth muscle cell through Gs adenylate cyclase (AC)—PKA signalling mechanism [57,58,59,60,61,62,63]. K^+^ efflux and thus hyperpolarization inactivates VDCC which results in smooth muscle relaxation and vasodilation due to decreased cytosolic Ca^2+^. Besides CGRP release, the trigeminal afferent C-fibres also release other vasoactive peptides including substance P, which all together increase vessel dilation and permeability and induce vascular inflammation by local release of nociceptive molecules (e.g., serotonin, bradykinin, histamine, and prostaglandins) [64].

### 5.2. The Trigeminal Ganglion

Experimental research revealed the expression of ASIC, BK_Ca_, K_ATP_, NMDAR, TRPA1, and TRPV1 at neuronal soma in the TG. These channels are inwardly streaming cation channels, except BK_Ca_ and K_ATP_, and their mutual activity determines action potential propagation, CGRP-release, and nociceptive information in the trigeminal pain pathway. Membrane depolarization upon opening of ASIC, TRPA1, and TRPV1 channels removes the voltage-dependent Mg^2+^-blockade in NMDARs and induces further membrane depolarization. In addition, membrane depolarization itself can activate BK_Ca_ and K_ATP_ channels. Ca^2+^/CaM complexes, formed by increased cytosolic Ca^2+^, activated Ca^2+^/CaM-stimulated AC (AC isotype 1 and 8), which in turn, activated several ion channels including NMDAR, causing enhanced pain perception through AC-cAMP-PKA signalling [65,66]. How BK_Ca_ and K_ATP_ channels are modulated by Ca^2+^/CaM-stimulated AC signalling is yet not clarified. Preclinical studies have demonstrated that opening of BK_Ca_ and K_ATP_ channels caused decreased neuronal activity, and K_ATP_ channels additionally inhibited neurotransmitter release [34,67,68,69,70]. By this means, BK_Ca_ and K_ATP_ channels located in TG could affect the release of vasoactive peptides such as CGRP and neurotransmitters such as glutamate. It should be noted that hyperpolarization-activated and cyclic nucleotide-gated (HCN) channels are present throughout the trigeminal neurons and drive Na^+^ into the cell in response to membrane hyperpolarization [71] and hereby support membrane depolarization. The exact purpose and effect of BK_Ca_ and K_ATP_ channels in TG and TNC needs to be investigated to understand the contribution of these channels in migraine nociception.

### 5.3. The Trigeminal Nucleus Caudalis

Preclinical data showed that stimulation of dural structures mediated a co-release of glutamate and CGRP in the TNC [72]. Moreover, CGRP has been shown to facilitate glutamate-driven neuronal nociception in mice [73]. Thus, it is expected that activation of CGRP receptors in central trigeminal Aδ-fiber terminals [74] might induce glutamate release and activation of NDMARs in TNC, and central trigeminal C-fibres may facilitate CGRP release. The post-synaptic membrane of TNC expresses NMDARs which induce activation of K_ATP_ and BK_Ca_ channels through depolarization and increased intracellular Ca^2+^ levels for the BK_Ca_ channel specifically. Collectively, K_ATP_ and BK_Ca_ channels are expressed at several levels of the trigeminal pain pathway and their activation seems to initiate cephalic nociception. Based on ion channel expression in the trigeminovascular system, the TG has the highest expression of ion channels, followed by the dural afferents, e.g., the peripheral trigeminal sensory nerve terminals. However, Iberiotoxin, a BK_Ca_ blocker, induced an increase in CGRP release from the TNC, which subsequently was attenuated by the BK_Ca_ channel opener, NS11021 [75]. This finding indicates a site-dependent effect of potassium channels. Potassium channel activation within the peripheral nervous system causes chemical (K^+^ efflux) and mechanical (vasodilation) activation of trigeminal afferents leading to cephalic nociception, whereas potassium channel activation within the central nervous system causes neuronal hyperpolarization and a decrease in neurotransmitter release. This site-dependent regulation of nociception should be taken into consideration for targeted therapy development in migraine.

### 5.4. The Relevance of Ion Channel Interplay

Thus far, the common approach is to singly investigate the role of ion channels by knock-out or by applying channel-specific modulators. The contribution of ion channels in the trigeminal pain pathway as a unity and their interplay has not been assessed deeply. Targeting a single ion channel among a diverse group of channels might probably not demonstrate a significant difference considering the great interplay of signalling. For instance, experimental studies investigating K_ATP_ channel agonist-induced CGRP-release and meningeal vasodilation concludes that the outcome is related to K_ATP_ channel activation. In this case, the interplay between the K_ATP_ channel and other channels expressed on the same location (e.g., dural afferents or TG) needs to be investigated. It is not surely known whether the K_ATP_ channel directly mediates CGRP release or indirectly through co-activation of nearby located ion channels, such as the NMDAR in TG. Another point regarding ion channel interplay is that an acidic environment triggers the opening of ASIC, TRPA1, and TRPV4 channels, which are all expressed in both dural afferents and TG. Again, whether one of these channels is dominant and controls the others, all three channels contribute equally, or whether there is an unidentified player are yet to be elucidated. To investigate the ion channel interaction, channels expressed on the same location could be 1) marked through immunohistochemistry, 2) blocked, or 3) knocked out during the examination of one particular ion channel.

### 5.5. Regulation of Ion Channel Expression

Preclinical data have confirmed the upregulation of some ion channels during neuroinflammation [76,77]. Thus, a disruption of the balance between ion channel expression (through secretory pathways) and channel internalization (through endocytosis) could be associated with neuronal hypersensitivity and increased neuronal firing, explaining some of the mechanisms behind the phases of migraine, including the aura and headache phases. Therefore, the question arises: can ion channel expression be downregulated? Ion channel internalization can be triggered by specific conditions, such as activation of certain receptors (e.g., GPCRs). In this context, ligand-dependent receptor activation triggers post-translational modification of ion channels (e.g., phosphorylation and ubiquination), which induces internalization [78]. It is known that activation of protein kinase C (PKC) inhibits BK_Ca_ and K_ATP_ channels in vascular smooth muscle cells, for instance, by angiotensin II, whose receptor is a GPCR. A significant aspect of this inhibition was reported in 2008, revealing that activation of PKC caused caveolin-dependent internalization of K_ATP_ channels (Kir6.1/SURB2 subtype), and a reduction in the number of K_ATP_ channels in smooth muscle plasma membrane was observed [79]. This finding supports that ion channels might be rapidly downregulated by internalization, and further research regarding downregulation of ion channel in the neuronal environment such as in trigeminal afferents, TG, and TNC would open a novel therapeutic mechanism in ion channel targeting.

### 5.6. Neuronal Hyperexcitability

Several findings indicated brain hyperexcitability during migraine aura and migraine pain [80]: (1) exaggerated CO_2_ reactivity [81], (2) hyperperfusion and abnormal cerebrovascular reactivity [82], (3) abnormal energy metabolism [83], and (4) low phosphocreatine, high adenosine 5′-diphosphate (ADP), and a low phosphorylation PCr:Pi ratio [84]. Brain hyperexcitability may be caused by low magnesium levels [85], mitochondrial abnormalities with abnormal phosphorylation of ADP, a dysfunction related to NO, and/or channelopathy [15,16,84]. Low magnesium increases the open probability of the NMDA receptor and results in the opening of calcium channels, increased intracellular Ca^2+^, and increased extracellular K^+^. A possible mitochondrial dysfunction with abnormal phosphorylation of ADP decreases the ADP/ATP ratio. The latter is essential to maintain intracellular functions including Ca^2+^ and K^+^ homeostasis. Potassium channels have been shown to exhibit activity within the inner mitochondrial membrane, including K_ATP_ (mitoK_ATP_) and BK_Ca_ (mitoBK_Ca_) channels [86,87]. They affect the integrity of mitochondrial inner membranes, leading to the regulation of energy-transducing processes and the synthesis of reactive oxygen species (ROS) [88,89]. In principle, all drugs (blockers and openers) acting on mitochondrial potassium channels have also been previously found to regulate plasma membrane potassium channels.

The fundamental question is how K_ATP_ and BK_Ca_ channels fit in the theory of migraine brain hyperexcitability. During neuronal hyperexcitability and according to the basic physiology of these channels, low ATP level might activate K_ATP_ channels and increased intracellular calcium might activate BK_Ca_ channels. Activation of these channels might, at least partly, explain increased extracellular K^+^. Now that activation of these channels causes hyperpolarization, the question becomes how direct activation of K_ATP_ (upon levcromakalim administration) and BK_Ca_ (upon MaxiPost administration) channels causes hyperexcitability. Potassium-channel-induced hyperpolarization activates cyclic nucleotide-gated cation channels (HCN channels) resulting in a generation of an inward current [90]. This notion is supported by the finding that K_ATP_ channel activation increased the firing rate of nigral dopaminergic neurons [91].

## 6. Other Ion Channels

### 6.1. Transient Receptor Potential Channels

Transient receptor potential (TRP) channels are Ca^2+^ and Na^+^ permeable cation channels, responsible for encoding and transducing different sensory stimuli including auditory, olfactory, thermal, and visual stimuli, and environmental irritants to nociceptive signalling [92,93]. Numerous studies implicated TRP channels in the pathophysiology of headache and suggested that this family might represent novel targets for headache therapeutics [94,95]. Mammalian TRP channels are composed of six transmembrane domains (S1–S6) with a pore domain (P) between the fifth and sixth domain. The TRP family is divided into six groups (TRPA, TRPC, TRPM, TRPML, TRPP, and TRPV) [96,97]. The interest in the involvement of TRP channels in migraine pathophysiology is mainly due to their expression on meningeal nociceptors, in particular TRPA1, TRPM8, TRPV1, and TRPV4 [98], and their role in CGRP release from sensory nerve endings upon activation [99,100].

The TRP Valinoid 1 (TRPV1) channel was one of the first TRP channels to be investigated, and it is expressed in small- and medium-sized neurons, mainly unmyelinated C-fibres or Aδ-fibers in trigeminal and dorsal root ganglion (DRG) neurons [57,100]. TRPV1 channels are mainly activated by capsaicin, noxious temperatures above 42 °C, and a variety of endogenous and exogenous compounds such as anandamide, endocannabinoids, and prostaglandins. Numerous studies have used capsaicin and TRPV1 antagonists to investigate the meningeal afferent and vascular function and suggested a solid role for TRPV1 in headache mechanisms [101]. A clinical study in 2014 demonstrated a significant increase in TRPV1 expression on periarterial nociceptive fibres of scalp arteries in individuals with chronic migraine compared with healthy controls [102]. Repeated 30-day administration of antimigraine drugs (eletriptan or indomethacin) in rats upregulated TRPV1 and TRPA1 in the TG, indicating the involvement of these channels in medication overuse headache [76]. Moreover, the relation between TRPV1 and CGRP release was examined by the administration of capsaicin and ethanol in animal studies, which were shown to promote neurogenic inflammation and CGRP-mediated dural vessel dilation [103,104]. Despite a suggestive role for TRPV1 in the migraine headache mechanism, the efficacy of TRPV1-antagonists in anti-migraine therapy is still uncertain.

The TRP Ankyrin 1 (TRPA1) channel is distinguished from other TRP channels by the presence of 14 ankyrin repeats in the N-terminus, linking cytoskeletal proteins to the channel directly. TRPA1 is a common pathway for a large number of pronociceptive agonists including environmental irritants such as cigarette smoke, umbellulone, acrolein, and reactive oxygen species [105]. In preclinical models, the application of TRPA1 agonists, mustard oil, and umbellulone, evoked TRPA1-like currents in approximately 42% and 38% of dural afferents, respectively, and resulted in meningeal vasodilation and CGRP release [59,60].

The role of the TRP melastatin 8 (TRPM8) channel in migraine was investigated after genome-wide association study (GWAS) analyses on three different groups of individuals with migraine. All three groups revealed a TRPM8 gene variant associated with increased susceptibility to migraine [106,107]. In the absence of other meningeal afferent stimuli, TRPM8 activation results in increased pain perception and vice versa when nearby afferents receive stimuli.

TRP valinoid 4 (TRPV4) is a Ca^2+^ and Mg^2+^ permeable cation channel that responds to a number of stimuli including changes in osmolarity, moderate heating, and lastly, 4α-PDD—a chemical compound classified as phorbol ester [108]. In addition, the channel is sensitive to mechanical forces imposed on the cell membrane [95]. TRPV4 is found in both meningeal nociceptors and the TG [109]. Since dural afferent nociceptors are mechanically sensitive, TRPV4 appears as a possible candidate for directly mediating the mechanosensitivity of dural afferent nociceptors. Activation of TRPV4 with hypotonic solutions and 4α-PDD within the meninges produced afferent nociceptive signalling and caused headache behavioural responses in rats [109], which were blocked by the TRPV4 antagonist RN1734. The relation between migraine headache and TRPV4 lies in the mechanosensitive activation of dural afferent nociceptors; a mechanical stimulation of TRPV4 followed by sudden changes in intracranial pressure (e.g., coughing, sneezing, standing/sitting, or exercising) increase the sensitivity of meningeal nociceptors and exacerbate migraine headache.

### 6.2. Acid-Sensing Ion Channels

In the early 1980s, acid-evoked currents were observed in neurons [110]. Approximately 20 years later, the ASIC responsible for the acid-evoked currents was cloned and identified [111,112]. Four ASIC genes (ASIC1, ASIC2, ASIC3, and ASIC4) and six ASIC subunits (ASIC1A, ASIC1B, ASIC2A, ASIC2B, ASIC3, and ASIC4) have been mapped. Three homo or hetero subunits combine into a trimeric channel complex with wide range of distinct properties [113]. The complex ASICs family are permeable to cations, primarily Na^+^ and to a lesser degree Ca^2+^, and are activated by extracellular acidosis and modulated by various factors including extracellular alkalosis [114,115]. Interestingly, pH sensitivity varies widely across ASIC subtypes to establish a representative range covering the physiological and pathophysiological alternation in pH. Upon activation, an inward current depolarizes the cell membrane and activates voltage-gated Na^+^ channels (VGSCs) and voltage-gated Ca^2+^ channels (VGSCs) resulting in NMDR receptor activation through the release of the Mg^2+^ blockade [113].

In brain neurons, ASIC1A is the dominant subunit found in the cell body, in dendrites, and in postsynaptic dendritic spines, indicating its role in synaptic physiology [113]. In the spinal cord, ASIC1A and ASIC2A levels were increased by peripheral inflammation, suggesting a role for ASICs in the central sensitization of pain [77]. At the peripheral sensory neurons terminal, mechanical stimuli as well as protons and other endogenous or exogenous chemicals are thought to activate several subtypes of ASICs. In the preclinical model, activation of ASIC3 triggered pain behaviours in wild-type but not in ASIC3-knockout mice. Furthermore, inhibition of ASIC1A and ASIC2A in the CNS and ASIC1B in the PNS reduces pain [116,117]. These findings highlight the possibility that the CNS and PNS use different combinations of ASIC subunits to mediate pain.

The ASIC family has also been suggested to play a part in epilepsy. Seizures reduce brain pH, and it is well established that acidosis inhibits seizure possibly because of feedback inhibition mediated by low pH at ASIC channels. Building on these observations, overexpressing ASIC1A in mice inhibited seizures and ASIC1A-knockout mice had prolonged chemoconvulsant-induced seizures without altering the seizure threshold [118]. Thus, ASIC1A emerged to be a novel target for treating epilepsy and status epilepticus. Epilepsy and migraine are common episodic neurological disorders with apparently shared pathological mechanisms. Comorbidity studies revealed that the prevalence of migraine in populations of individuals with epilepsy is approximately twice that in the normal population. More importantly, the introduction of antiseizure medications, particularly the second-generation, has been advantageous for migraine patients, and several anti-epileptics including valproate and topiramate are FDA approved for the prevention of migraine.

The induction of tissue hypoxia and disruption by CSD, involvement of ASIC channels in pain modulation and seizure, comorbidity data between migraine and epilepsy, and that a number of anti-epileptic agents are proven preventive treatments in migraine, implicate ASIC channels in migraine pathogenesis [119]. The antihypertensive ASIC1 inhibitor amiloride is approved for use in humans, and a few small translational experiments have demonstrated its potential for reducing cutaneous pain and migraine. Taken together, available data so far offer a strong indication that the ASIC1 subunit may offer a therapeutic target in migraine.

## 7. Concluding Remarks

Migraine is a complex disease involving various pathological mechanisms. Meningeal arteries with trigeminal afferents denoted as the TVS is the anatomical substrate for migraine pain. Potassium channels, particularly K_ATP_ and BK_Ca_ channels, are expressed at several levels of the TVS where they exert a key role in migraine attack initiation, propagation, and duration. Endogenous signalling molecules involved in migraine including CGRP and PACAPs are dependent on potassium channel activation. Direct activation of K_ATP_ or BK_Ca_ channels dilated cranial arteries and induced headache in healthy volunteers and migraine attacks in individuals with migraine. Several aspects of potassium channel involvement in migraine pathogenesis remain unrevealed including the exact anatomical location, the specific subunits expressed in the TVS, and the interplay between ion channels. Moreover, clinical-approved selective antagonists are required to further elucidate their implication.

## Figures and Tables

**Figure 1 pharmaceuticals-16-00438-f001:**
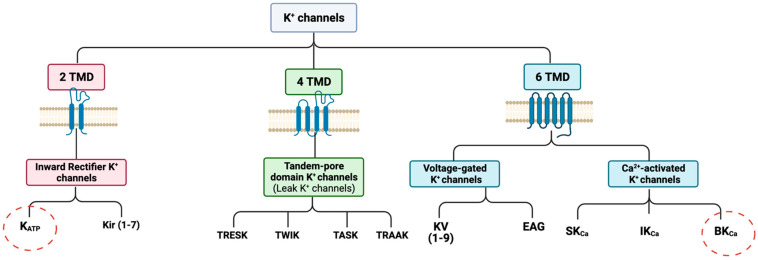
**Classification of potassium (K^+^) channel family.** The K^+^ channels are structurally divided into three subclasses based on the number of transmembrane segments. K^+^ channels with two transmembrane domains (2 TMD) are known as Inward Rectifier K^+^ channels, to which K_ATP_ and Kir channels belong. Tandem-pore domain K^+^ channels, also called ‘leak K^+^ channels’, consist of four transmembrane domains, whereas voltage-gated and calcium-activated K^+^ channels are composed of six transmembrane domains (6 TMD). Furthermore, the calcium-activated K^+^ channels are named according to their calcium conductivity. **K_ATP_** = ATP-sensitive potassium channel; **Kir** = inward rectifying K^+^ channel; **TWIK** = tandem weak inward rectifying K^+^ channel; **TRESK** = TWIK-related spinal cord K^+^ channel; **TASK** = TWIK-related acid-sensitive K^+^ channel; **TRAAK** = TWIK-related arachidonic acid-activated K^+^ channel; **KV** = voltage-gated K^+^ channel; **EAG** = ether-a-go-go K^+^ channel; **SK_Ca_** = small conductance calcium-activated K^+^ channel; **IK_Ca_** = intermediate conductance calcium-activated K^+^ channel; and **BK_Ca_** = big conductance calcium-activated K^+^ channel.

**Figure 2 pharmaceuticals-16-00438-f002:**
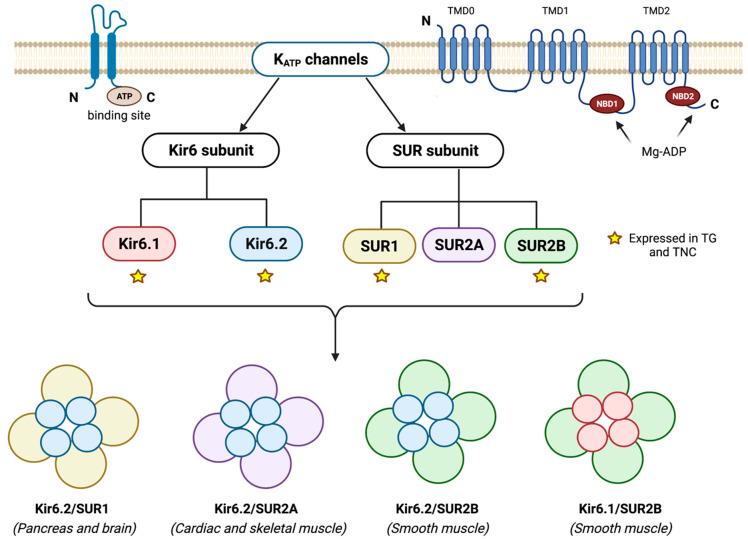
**Structure of the K_ATP_ channel.** The tetrameric K_ATP_ channel complex is assembled by four Kir6 and four SUR subunits. The Kir6 subunit is a 2 TMD with an ATP-binding site, whereas the SUR subunit consists of 3 components (TMD0, TMD1, and TMD2) with 5–6 transmembrane domains in each. The SUR subunit includes the nucleotide-binding domain (NBD) for Mg-ADP. The distinctive subtypes in each subunit give rise to different combinations and properties of the K_ATP_ channels. Kir6.1, Kir6.2, SUR1, and SUR2B are expressed in TG and TNC. The occurrence of these combinations is presented. **Kir6** = inwardly rectifying K^+^ channel; **SUR** = sulphonyl urea receptor; **TG** = trigeminal ganglion; **TNC** = trigeminal nucleus caudalis.

**Figure 3 pharmaceuticals-16-00438-f003:**
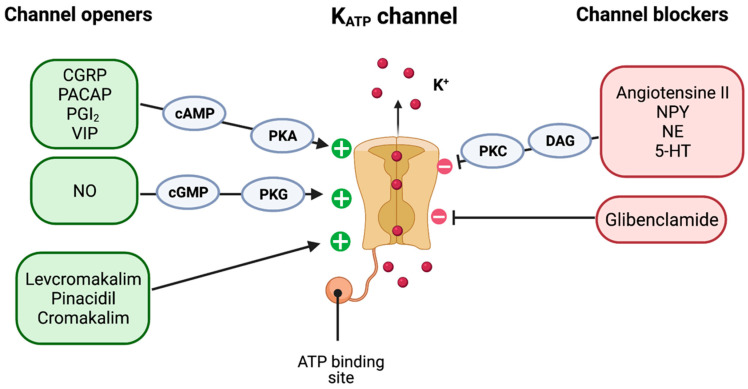
**K_ATP_ channel openers and blockers.** K_ATP_ channels on smooth muscle cells can be opened by endogenous vasoactive compounds such as CGRP, PACAP, VIP, and NO, and moreover, directly opened by synthetic channel openers (e.g., levcromakalim and pinacidil). Conversely, a DAG-PKC phosphorylation-dependent mechanism is seen in endogenous channel blockers, such as angiotensin II and NE, whereas the synthetic channel blocker glibenclamide directly inhibits the K_ATP_ opening and smooth muscle relaxation. **5-HT** = 5-hydroxytryptamine; **NE** = norepinephrine; **NO** = nitric oxide; NPY = neuropeptide Y; **PGI_2_ =** prostaglandin I_2_; and **VIP** = vasoactive intestinal peptide.

**Figure 4 pharmaceuticals-16-00438-f004:**
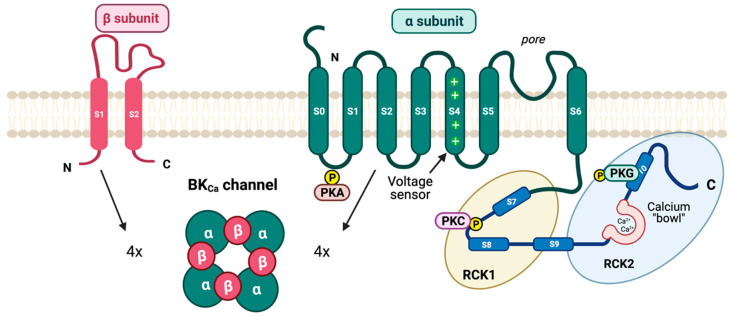
**The structure of BK_Ca_ channel.** The BK_Ca_ is a tetrameric channel complex assembled by four pore-forming α-subunits and four regulatory β-subunits. The α-subunit includes transmembrane domains (S0–S6) at the *N*-terminus and intracellular domains (S7–S10) at the *C*-terminus. The pore is formed between S5 and S6, whereas the S4 segment constitutes a voltage sensor. The phosphorylation site for PKA is found in the transmembrane domains, while the sites for PKC and PKG are located in the intracellular domains. In addition, the S7–S10 segments are associated with a regulatory potassium conductance domain (RCK1 and RCK2).

**Figure 5 pharmaceuticals-16-00438-f005:**
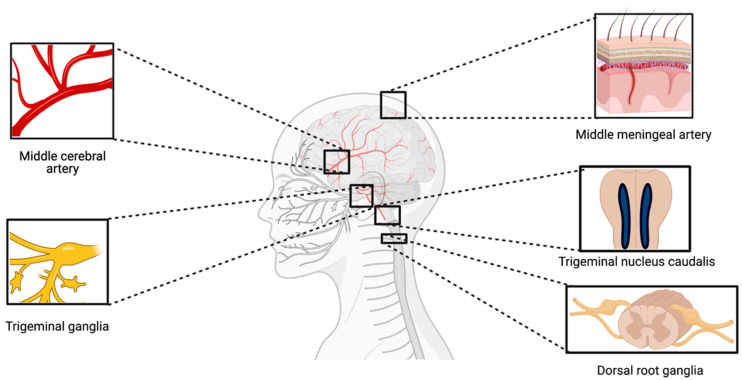
Distribution of BK_Ca_ channels in the trigeminovascular system.

**Figure 6 pharmaceuticals-16-00438-f006:**
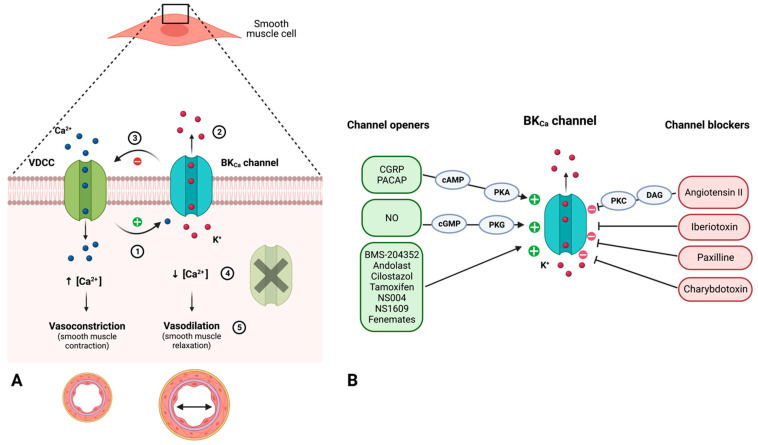
(**A**) **The regulation of vascular tonus by BK_Ca_ channels.** (**B**) **BK_Ca_ channel openers and blockers**. (**A**): Opening of voltage dependent calcium channels (VDCC) causes influx of Ca^2+^ ions and membrane depolarization, which in turn, opens the BK_Ca_ channel (1). Activation of BK_Ca_ causes efflux of K^+^ ions and subsequent membrane hyperpolarization (2), inducing negative feedback on the depolarization-dependent Ca^2+^ influx by VDCC (3). Decrease in cytosolic Ca^2+^ concentrations due to inactivated VDCC results in smooth muscle relaxation and vasodilation (4). (**B**): NO and endogenous vasoactive peptides such as CGRP and PACAP act as BK_Ca_ openers through PKG and PKA phosphorylation, respectively, on vascular smooth muscle cells. The present figure illustrates several synthetic direct channel openers such as BMS-204354 (MaxiPost) and NS1609. Among the presented BK_Ca_ channel blockers, the endogenous vasoconstrictor, angiotensin II, acts through the DAG and PKC-phosphorylation mechanism, whereas other synthetic channels directly block the opening of BK_Ca_. **CGRP** = calcitonin gene-related peptide; **PACAP** = pituitary adenylate-cyclase activating peptide; **NO** = nitric oxide; **cAMP** = cyclic adenosine monophosphate; **cGMP** = cyclic guanosine monophosphate; **DAG** = diacylglycerol; and **PKA/C/G** = protein kinase A/C/G.

**Figure 7 pharmaceuticals-16-00438-f007:**
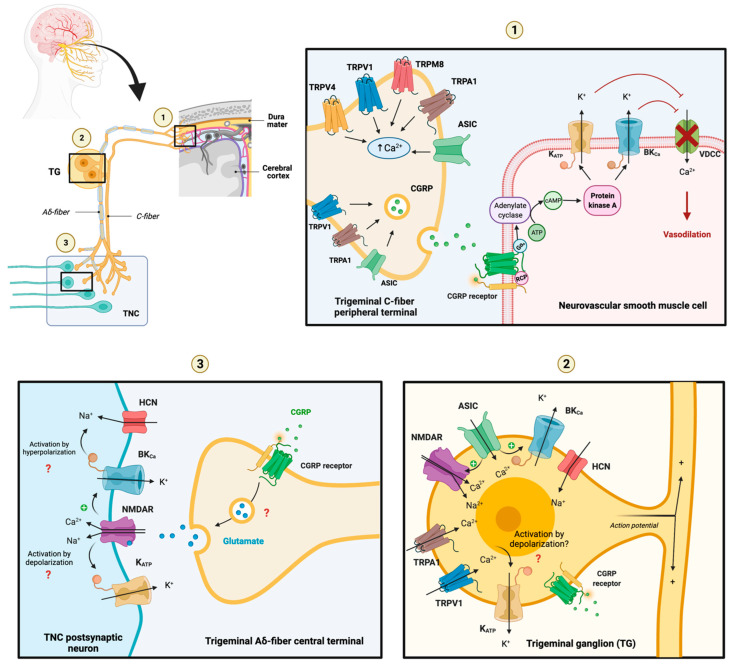
**Molecular interplay between ion channels in trigeminovascular system.** (**1**) Peripheral terminal of trigeminal C-fibre and neurovascular smooth muscle cell. (**2**) The trigeminal ganglion (TG). (**3**) Central terminal of trigeminal Aδ-fiber and neuron within the trigeminal nucleus caudalis (TNC). Nociceptive signals from the meninges and other cranial tissues including blood vessels reach multiple cortical areas through the trigeminal pain pathway consisting of peripheral trigeminovascular neurons in the TG, central trigeminovascular neurons in the TNC, and thalamic neurons (not shown). The dural and pial blood vessels are innervated by sensory and autonomic nerves that express vasoactive neuropeptides including CGRP (the most abundant in sensory neurons co-expressed with PACAP), substance P, and VIP (primarily found in autonomic neurons). Knowledge about the response properties of leptomeningeal sensory and autonomic nerves and their activation during migraine headache is limited. It is believed that local sterile meningeal inflammation mediates the prolonged activation and sensitization of meningeal nerves leading to migraine headache. However, the origin of such neurogenic inflammation remains elusive. Upon inflammation, action potentials from activated sensory fibres are conducted antidromically and invade peripheral end branches resulting in the release of vasoactive substances. Activation of K_ATP_ and BK_Ca_ channels in neurovascular smooth muscle cells causes K^+^ outflow (chemically induced sensitization) and vasodilation (mechanically induced sensitization) [49]. Perivascular sensory afferents are hereby further sensitized. Whether and by what mechanisms ion channels expressed in the TG affect signal transduction is yet to be elucidated. Action potentials reach the central terminal of the trigeminal Aδ-fiber and cause a release of glutamate leading to activation of neurons within the TNC. Glibenclamide, a non-selective blocker of K_ATP_ channels, failed to inhibit cephalic vasodilation and headache [38,39,40,41,56]. **ASIC** = acid-sensing ion channel; **BK_Ca_** = big conductance calcium-activated K^+^ channel; **CGRP** = calcitonin-gene related peptide; **HCN** = hyperpolarization-activated and cyclic-nucleotide-gated channel; **K_ATP_** = ATP-sensitive K^+^ channel; **NMDAR** = *N*-methyl d-aspartate receptor; PACAP = pituitary adenylate cyclase-activating polypeptide; **RCP** = Receptor component protein; **TG** = trigeminal ganglion; **TRP** = transient receptor potential; **TNC** = trigeminal nucleus caudalis; **TGVS** = trigeminovascular system; **VDCC** = voltage-dependent Ca^2+^ channel; and **VIP** = vasoactive intestinal polypeptide.

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
