# Peer review of "Involvement of Potassium Channel Signalling in Migraine Pathophysiology"

_pharmaceuticals, 2023, doi:10.3390/ph16030438_

Round 1

Reviewer 1 Report

The present review outlines the molecular structure and physiological function of KATP and BKCa channels highlights recent mechanistic insights into their role in migraine pathophysiology and discusses the potential targeting of these channels in the treatment of headaches and migraine.

It is a large, well-documented study, the topic original or relevant in the field,  based on a narrative search of the PubMed database regarding potassium channels and migraine, I

I recommend the author improve the introduction by providing more Information about migraine and ion channels in the pathogenesis of migraine, and also I recommend improving English grammar.

The conclusions are insufficient, substantial additions should be made.

Regarding the references, I recommend replacing manuscripts older than five years with up-to-date ones.

Author Response

Reviewer #1

Comments for the Author

  1. The present review outlines the molecular structure and physiological function of KATP and BKCa channels highlights recent mechanistic insights into their role in migraine pathophysiology and discusses the potential targeting of these channels in the treatment of headaches and migraine. It is a large, well-documented study,the topic original or relevant in the field, based on a narrative search of the PubMed database regarding potassium channels and migraine, I recommend the author improve the introduction by providing more information about migraine and ion channels in the pathogenesis of migraine, and also I recommend improving English grammar. The conclusions are insufficient, substantial additions should be made. Regarding the references, I recommend replacing manuscripts older than five years with up-to-date ones.

Response: Thank you for your comments. The above-mentioned concerns are addressed in page 1 and 14.

Reviewer 2 Report

This is a classic overview showing the role of KATP and BKCa potassium channels in the pathophysiology of migraine. The author has experience with these channels, which is important for writing a good review. However, I will note important critical points that prevent the paper from being accepted for publication at this stage.

1. It is unacceptable to use only the PubMed database to search for sources. Many good research papers are not indexed into this database for various reasons. Therefore, the author needs to conduct a more thorough analysis of the literature data, using other databases as well. This will provide a complete picture of the involvement of channels in the pathophysiology of migraine.

2. The title of the work does not reflect its content. The author himself indicates the presence of many potassium channels, but describes mainly KATP and BKCa. This must be taken into account in the title.

3. The author needs to show that the channels described are also found in organelles, this is important. In particular, mitochondrial potassium channels are also involved in the pathophysiology of migraine.

4. Fig. 4b. NS1619, not NS619. Please check the figures carefully.

5. The list of references is not made according to the rules of the journal.

Author Response

Reviewer #2

Comments for the Author

This is a classic overview showing the role of KATP and BKCa potassium channels in the pathophysiology of migraine. The author has experience with these channels, which is important for writing a good review. However, I will note important critical points that prevent the paper from being accepted for publication at this stage.

  1. It is unacceptable to use only the PubMed database to search for sources. Many good research papers are not indexed into this database for various reasons. Therefore, the author needs to conduct a more thorough analysis of the literature data, using other databases as well. This will provide a complete picture of the involvement of channels in the pathophysiology of migraine.

Response: Thank you for your comment. Yes, it is a limitation to only use the PubMed database. However, this is a narrative review. I agree that the idea of a systematic review is interesting and definitely worth the time and effort it may take to address.

  1. The title of the work does not reflect its content. The author himself indicates the presence of many potassium channels but describes mainly KATP and BKCa. This must be taken into account in the title.

Response: Thank you for your comment. The present review includes two of the three subclasses of potassium channels. The third class is leak potassium channels. Therefore, the present title appears to be very propriate.   

  1. The author needs to show that the channels described are also found in organelles, this is important. In particular, mitochondrial potassium channels are also involved in the pathophysiology of migraine.

Response: Thank you for your comment. A new section is added to address this in page 12.

  1. Fig. 4b. NS1619, not NS619. Please check the figures carefully.

Response: Thank you for your comment. These are corrected.  

  1. The list of references is not made according to the rules of the journal.

Response: Corrected.  

Round 2

Reviewer 2 Report

The authors took into account some of my comments.